# Near-infrared STED nanoscopy with an engineered bacterial phytochrome

Maria Kamper[1], Haisen Ta[1], Nickels A. Jensen[1], Stefan W. Hell[1] & Stefan Jakobs [1,2]

The near infrared (NIR) optical window between the cutoff for hemoglobin absorption at 650 nm and the onset of increased water absorption at 900 nm is an attractive, yet largely unexplored, spectral regime for diffraction-unlimited super-resolution fluorescence microscopy (nanoscopy). We developed the NIR fluorescent protein SNIFP, a bright and photostable bacteriophytochrome, and demonstrate its use as a fusion tag in live-cell microscopy and STED nanoscopy. We further demonstrate dual color red-confocal/NIR-STED imaging by co-expressing SNIFP with a conventional red fluorescent protein.

---

[1] Department of NanoBiophotonics, Max Planck Institute for Biophysical Chemistry, Göttingen, Germany. [2] Department of Neurology, University Medical Center Göttingen, Göttingen, Germany. These authors contributed equally: Maria Kamper, Haisen Ta, Nickels A. Jensen. Correspondence and requests for materials should be addressed to S.J. (email: sjakobs@gwdg.de)

Within the near-infrared (NIR) optical window at around 650–900 nm, light scattering, autofluorescence and light absorbance is strongly reduced in mammalian cells and tissues[1]. Therefore, this spectral region is preferable for deep-tissue imaging[2]. Phototoxic effects, even at high irradiation intensities, are generally alleviated at long wavelengths[3], rendering the NIR regime attractive for live-cell microscopy and particularly for live-cell diffraction-unlimited super-resolution (nanoscopy) applications that generally require the application of higher light doses[4].

Indeed, substantial efforts have been undertaken to generate organic dyes that can be used for live-cell super-resolution microscopy in the NIR regime[5,6]. Likewise, fluorescent proteins (FPs) are routinely used for live-cell nanoscopy and extensive, albeit unsuccessful, efforts have been undertaken to generate FPs of the green fluorescent protein (GFP) family that are excited within the NIR optical window. To date, no GFP-like FP with an excitation maximum above 611 nm (TagRFP657[7] and E2-Crimson[8]) or an emission maximum above 686 nm (mNeptune681-Q159C[9]) has been reported[10]. Far-red fluorescent proteins (emission at around 650–670 nm) including TagRFP657[7], mNeptune2[11,12], mGarnet[13], and most recently mGarnet2[14] were used for stimulated emission depletion (STED) nanoscopy. However, the excitation wavelengths used were below 650 nm and thus outside of the NIR optical window.

Recently, NIR fluorescent proteins based on phytochromes, whose excitation and emission maxima are within the NIR optical window, have been developed and applied in fluorescence microscopy[15,16]. Phytochromes are found in bacteria, cyanobacteria, fungi, algae, and plants, but not in mammals. They rely on linear tetrapyrrole molecules, such as phycocyanobilin, phycoerythrin, or biliverdin, as external chromophores. Most phytochromes share a structurally conserved photosensory core module (PCM) of 55–58 kDa that is composed of a PAS (Per-ARNT-Sim), a GAF (cGMP phosphodiesterase-adenylate cyclase-FhlA), and a PHY (phytochrome-specific) domain which are connected by α-helical linkers[16]. Of the phytochromes, the bacterial phytochromes stand out, because they utilize the far-red absorbing biliverdin as a chromophore. As a product of the heme degradation pathway, biliverdin is ubiquitous in many eukaryotic organisms, and in addition, it is readily taken up by mammalian cells when exogenously applied. Several NIR FPs were engineered from different bacteriophytochromes[15,16]. Many of these NIR FPs are dimers and exhibit relatively low fluorescence brightnesses in mammalian cells, limiting their suitability for imaging. To date, none of these proteins have been used for nanoscopy.

In this study, we generated the new bacteriophytochrome-based, bright and photostable NIR FP SNIFP (STED Near Infrared Fluorescent Protein) that proved to be a suitable fusion tag when expressed in human cells. We demonstrate the use of SNIFP for live cell NIR-STED (NIR-STimulated Emission Depletion) nanoscopy, with all three wavelengths (excitation, emission, and STED) in the NIR optical window.

## Results

**Generation of the NIR fluorescent protein SNIFP.** As a starting scaffold, we chose the PCM of the *Deinococcus radiodurans* bacteriophytochrome (DR_A0050). The NIR FPs WiPhy[17], WiPhy2[18], IFP1.4[19], and IFP2.0[20] are variants of the truncated PCM of this bacteriophytochrome, containing only the PAS and the GAF domains. We revisited the question on the most suitable truncation or extension of the PCM to generate a NIR FP usable for live-cell imaging. To this end, we first introduced the monomerization mutations F145S, L311E, and L314E[17], and codon optimized the PCM sequence for expression in human

cells. We generated four different variants (W1–W4), with W1 being the longest (530 amino acids (aa)) and W4 the shortest variant (321 aa) (Supplementary Fig. 1; Supplementary Table 1).

Fluorescence activated cell sorting (FACS) revealed that the shortest variant W4 comprising only the PAS and the GAF domain, resulted in the highest fluorescence signal in *Escherichia coli* cells (Supplementary Table 1) and we thus continued to improve this variant. Guided by crystal structures of the PCM (PDB 4O0P and 4O01)[21] and previous mutational analysis[22], we chose a number of positions in the chromophore binding pocket for saturation mutagenesis (among others positions 206, 207, 208, 209, 216, 263, and 270) and combined it with polymerase chain reaction (PCR)-mediated random mutagenesis. The plasmid libraries were expressed in *E. coli* cells and screened in several consecutive rounds for fluorescence brightness by FACS analysis and automated microscopy[23].

We identified three mutations (D207L, Y263F, and G270R) that increased the fluorescence brightness of W4 in *E. coli* cells further. The positions 207 and 263 have been repeatedly identified as important for the fluorescence properties of bacteriophytochromes[17,22,24], whereas the position 270 has not been discussed. Gly270 is located between the α-helix 7 and the β-sheet 10 of the GAF-domain. The substitution of the glycine by the larger arginine is likely to influence the positioning of the highly conserved His260 and consequently the network of hydrogen bonds in the chromophore surrounding. To evaluate the influence of the mutation G270R, we compared four W4 variants, namely W4-Y263F (W4.20), W4-Y263F, G270R (W4.33), W4-Y263F, D207L (W4.34), and W4-Y263F, D207L, G270R (W4.35). For all four variants, the excitation (697–705 nm) and emission (720–723 nm) maxima are within the near infrared window (Supplementary Table 2; Supplementary Fig. 2), and they behave as monomers on semi-native polyacrylamide gels (Supplementary Fig. 3a). In *E. coli* cells, W4.34 and W4.35 were the brightest variants and showed a higher stability at pH values above 6.5 (Supplementary Table 2; Supplementary Fig. 3b). W4.35, which differs from W4.20 at one position (D207L), exhibited the highest extinction coefficient ($\sim$149,200 $M^{-1} cm^{-1}$) and a slightly higher cellular brightness in mammalian cells (Supplementary Table 2). Because W4.35 proved to be suitable for STED nanoscopy (see below), we named this protein SNIFP (STED Near Infrared Fluorescent Protein).

**SNIFP as a fusion tag in live cell imaging.** In order to evaluate the usability of SNIFP for live cell imaging, we fused it to the centromere protein C (SNIFP-CENPC) (Fig. 1a), to a core histone (SNIFP-HIST1H2bn) (Fig. 1b), to a subunit of the nuclear pore (SNIFP-NUP50) (Fig. 1c), to caveolin 1 (CAV1-SNIFP) (Fig. 1d), to keratin (KRT18-SNIFP) (Fig. 1e), to vimentin (VIM-SNIFP) (Fig. 1f), to a microtubule associated protein (SNIFP-MAP2) (Fig. 1g), and targeted it to the endoplasmic recticulum (ER) (Fig. 1h) and to the peroxisomes (Fig. 1i). These fusion proteins could be readily imaged in living human cells using a confocal microscope with a standard 633 nm excitation line (Fig. 1a–i), demonstrating that SNIFP is a suitable probe for far-red live cell microscopy. However, this excitation wavelength, which is the most red-shifted laser line in most current commercial confocal instruments, is outside the NIR window. Since it is shifted by $\sim$60 nm to the blue compared to the excitation maximum of SNIFP, it is also not optimal for its excitation. To fully benefit from the spectral properties of SNIFP, we employed an excitation laser line of 676 nm in a dedicated NIR confocal microscope. Thereby, we could image more than a thousand consecutive images of living cells expressing VIM-SNIFP (Fig. 1j, k).

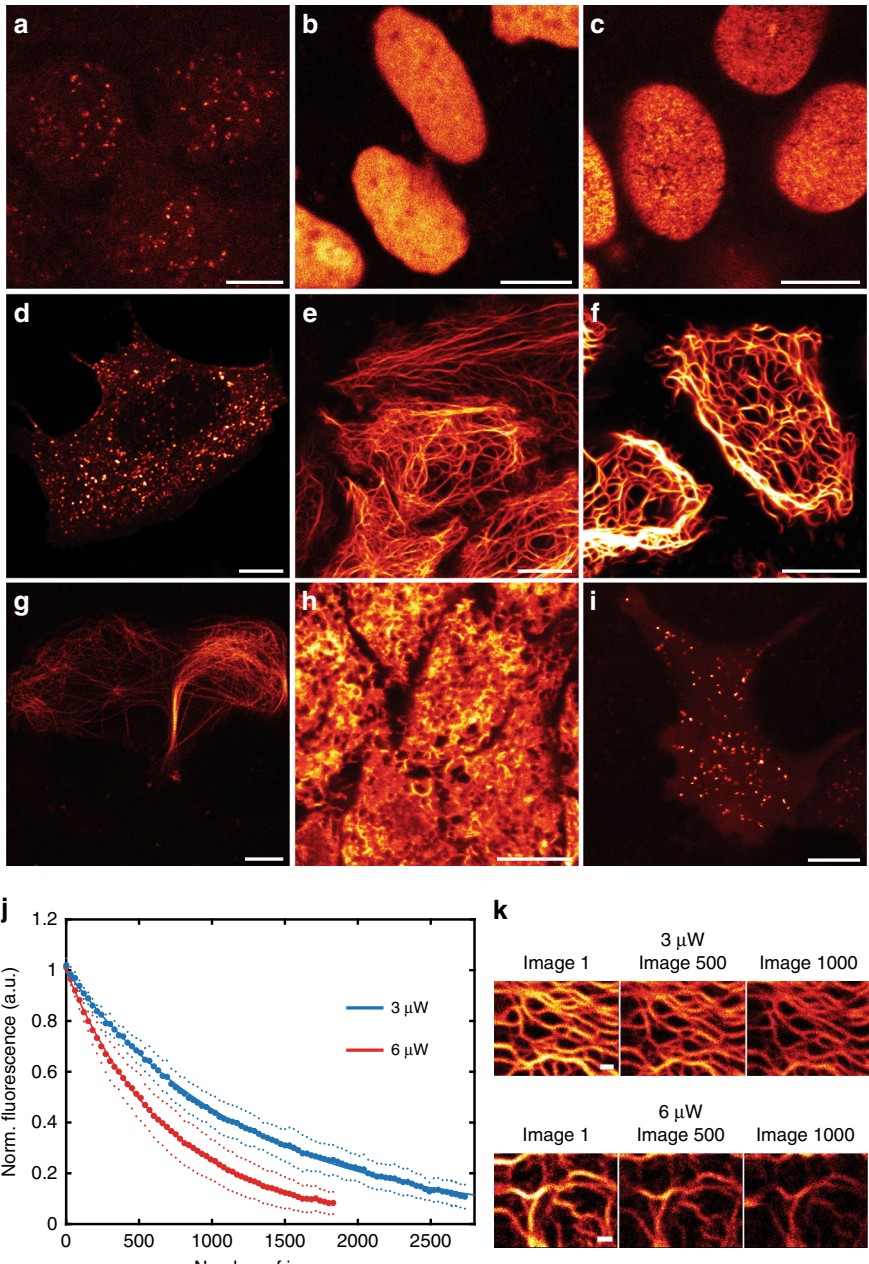

**Fig. 1** Confocal recordings of SNIFP in living HeLa and U2OS cells. **a–i** Images of SNIFP targeted to different subcellular structures. Fluorescence was excited at 633 nm. **a** Kinetochore, SNIFP-CENPC, **b** nucleus, SNIFP-HIST1H2bn, **c** nuclear pore, SNIFP-NUP50, **d** caveolae, CAV1-SNIFP, **e** keratin, KRT18-SNIFP, **f** vimentin, VIM-SNIFP, **g** microtubule associated protein, SNIFP-MAP2, **h** ER, ER-SNIFP, and **i** peroxisomes, SNIFP-PTS. **j–k** Bleaching of SNIFP in living human cells at 676 nm irradiation. **j** Confocal images of living Hela cells expressing VIM-SNIFP were continuously recorded at the same site using two different light intensities (3 or 6 µW, as indicated, measured in the back aperture of the objective lens). Each data point (large dots) represents the average of five measurements (blue, 3 µW) or six measurements (red, 6 µW). Small dots indicate the standard deviation. Only every 30th data point is displayed for better visualization. Solid line: single exponential fit to the data. The characteristic decay times (decay to 1/e of the initial signal) were 1280 ± 225 images (at 3 µw) and 726 ± 167 images (at 6 µW). **k** Representative confocal images, recorded with the indicated light intensities. Imaging parameters are listed in Supplementary Table 3. **a–k** 25 µM biliverdin was added to the medium ~2 h before imaging. Scale bars: 10 µm (**a–i**) and 1 µm (**k**)

**STED super-resolution microscopy with SNIFP**. Encouraged by the superior photostability of SNIFP upon confocal imaging, we next explored the suitability of SNIFP for STED nanoscopy[4,25]. To identify the optimal STED wavelength, we first determined the fluorescence signal induced by anti-Stokes excitation at different STED wavelengths (820–870 nm, provided by a tunable pulsed (80 MHz) Ti:Sapphire laser) on VIM-SNIFP in living cells. We found the wavelength of 860 nm to be optimal for STED nanoscopy of SNIFP, because this wavelength combines efficient

stimulated emission with a low level of anti-Stokes excitation (Supplementary Fig. 4).

We recorded STED images of living human HeLa cells expressing VIM-SNIFP or SNIPF-NUP50, demonstrating a clear resolution improvement compared to conventional confocal microscopy (Fig. 2a–c). All images display raw data. To quantify the resolution improvement, we determined the FWHM (Full Width at Half Maximum) of three neighboring averaged intensity profiles across the vimentin filaments. In the confocal case, the

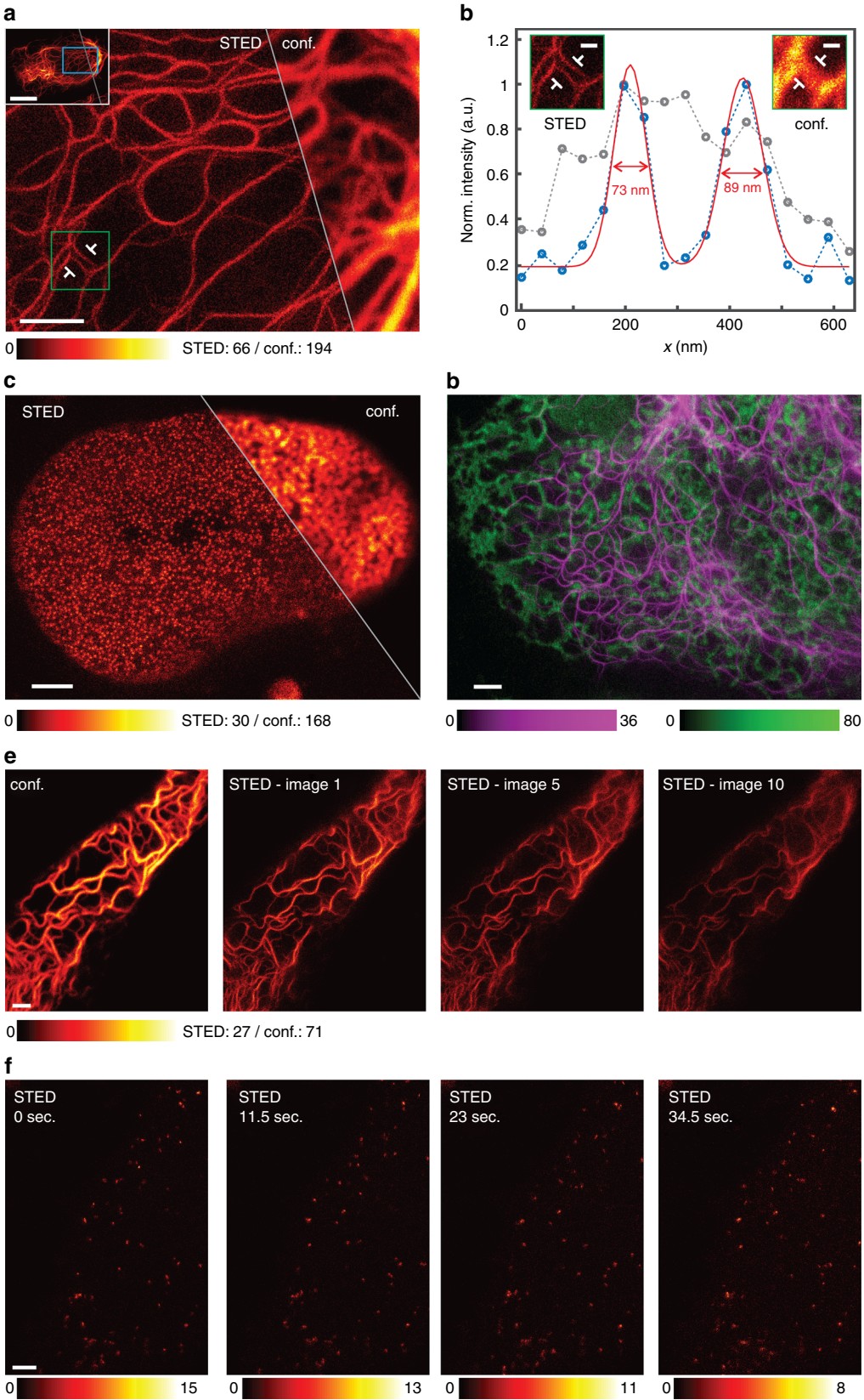

FWHM was measured as ~280 nm (Supplementary Figs. 5 and 6), which is close to the theoretical confocal resolution of 260 nm. In the STED recordings, the FWHM values were consistently around 80 nm (Fig. 2b; Supplementary Figs. 5 and 6), which corresponds to a ~3.5-fold improvement in resolution. We could record more than ten consecutive STED images, which is comparable to the number of recordings achievable with the far-red fluorescent protein mGarnet2 at a similar resolution using shorter wavelengths (Fig. 2e; Supplementary Fig. 7). SNIFP targeted peroxisomes exhibited vivid inner-cellular movements

**Fig. 2** NIR STED nanoscopy of living human cells expressing SNIFP fusion proteins. **a** STED nanoscopy of a HeLa cell expressing VIM-SNIFP. Large image: magnification of the region indicated in the inset showing the STED image of a whole cell (see also Supplementary Fig. 5). Left: STED nanoscopy, right: confocal microscopy. **b** Line profiles taken at the position indicated in **a**. The line width for averaging was 90 nm (three pixels). Blue circles: STED; gray circles: confocal; the averaged data were fitted with a Gaussian function (solid line). The FWHM values were determined on the fitted function. For more line profiles see Supplementary Figs. 5 and 6. **c** Live cell recording of the nucleus of a HeLa cell expressing SNIFP-NUP50. Left: STED nanoscopy, right: confocal microscopy. **d** Red confocal/NIR STED dual color imaging. VIM-SNIFP (magenta) was recorded in the STED mode, and the ER targeted mCherry (green) in the confocal mode. **e**, **f** Photostability of SNIFP during STED imaging. **e** Thirty consecutive STED images (recordings were started every 11 s) of a living Hela cell expressing Vim-SNIFP. Shown is a confocal image and three STED images, as indicated. (see Supplementary Fig. 7 for photostability analysis and comparison with mGarnet2). **f** Consecutive STED images recorded at the same site of a living HeLa cell expressing SNIFP-PTS to target the peroxisomes (recordings were started every 10.5 s). Shown are the first four images. **a**–**f** 25 μM biliverdin was added to the medium ~2 h before imaging. All images display raw data. Imaging parameters are listed in Supplementary Table 3. Scale bars: 10 μm (inset **a**), 2 μm (**a**, **c**–**f**), and 400 nm (**b**)

when imaged in the STED mode (Fig. 2f; Supplementary Fig. 8). As we added 25 μM biliverdin to the growth medium, we next ask the question if imaging is also possible without additional biliverdin. We found that both confocal as well as STED imaging is possible, albeit at a reduced signal-to-noise ratio (Supplementary Fig. 9). We conclude that addition of biliverdin is beneficial.

**Dual color red-confocal/NIR-STED.** Since SNIFP covers the NIR spectral regime, but leaves the spectrum below 676 nm untouched, SNIFP can be readily teamed with conventional red fluorescent proteins to enable red/NIR dual color imaging. To evaluate this option, we targeted the red fluorescent protein mCherry[26] having an excitation maximum at 587 nm, to the ER and co-expressed VIM-SNIFP. This allowed imaging the mCherry signal in the confocal, and the SNIFP signal in the super-resolved STED mode (Fig. 2d), demonstrating live cell imaging with two fluorophores emitting above 600 nm. Once additional suitable NIR fluorescent proteins are available that can be discriminated by, for example, the fluorescence lifetime or the excitation wavelength, also dual color NIR STED nanoscopy in living cells should be readily possible.

## Discussion

In conclusion, we engineered the bright and photostable bacteriophytochrome SNIFP and demonstrate its use for STED nanoscopy in the NIR spectral regime. As the NIR regime is particularly attractive for imaging living cells and for focusing deep into tissue, this is expected to pave the way towards multicolor live-cell deep tissue NIR nanoscopy.

## Methods

**Constructs for bacterial expression and mutagenesis.** For expression in *E. coli* cells, the respective coding sequences were cloned into a pBad/HisB expression plasmid (Addgene plasmid #14892). The sequences were PCR amplified, digested (*Eco*RI and *Sal*I) and ligated into the digested (*Eco*RI and *Xho*I) pBad/HisB vector.

PCR-based random error-prone mutagenesis, site-directed mutagenesis and multiple site mutagenesis were performed according to standard protocols[27,28].

**Protein expression and purification.** For spectroscopic measurements in cells, proteins were expressed at 37 °C in *E. coli* strain BL21-AI™ (Invitrogen, Carlsbad, CA, USA) transformed with the plasmid pWA23h[2], enabling rhamnose-induced biliverdin synthesis. For cultivation on agar plates and in liquid culture, the LB medium was supplemented with antibiotics (ampicillin, 50 μg/ml and kanamycin, 50 μg/ml) for selection. Arabinose (0.02%) and rhamnose (0.2% in liquid culture, 0.02% in agar plates) were used for induction of expression. 50 μM FeCl₃ and 100 μM δ-aminolevulinic acid (ALA) (Sigma-Aldrich, St. Louis, MO, USA) were added to the medium[2] to ensure biliverdin availability.

For protein purification, proteins were expressed in BL21-AI™ cells (Invitrogen) without the pWA23h plasmid. After expression for 20 h at 37 °C followed by 1 h at room temperature, purification was performed by Ni-NTA affinity chromatography (His SpinTrap Kit, GE Healthcare, Little Chalfont, BKM, GB) according to the manufacturer's instructions with a 30 min binding step. After purification, the protein concentration was determined using the BioRad (Hercules, CA, USA) protein assay and subsequently the solution was supersaturated 3-fold with biliverdin. With repeated washing steps using Vivaspin 500 colums (Sartorius, Göttingen, DE) unbound biliverdin was washed out and the proteins were taken up in 100 mM Tris-HCl, 150 mM NaCl, pH 7.5.

**Constructs for expression in mammalian cells.** To target SNIFP to MAP2, Hist1H2BN or the peroxisomes, the coding sequence of SNIFP was amplified using the primers 5′ATCCGCTAGCGCTAATGTCCCGTGACCCTCTC3′ and 5′CACTCGAGATCTGAGTCCGGATTCTTTGACTTGCACCTGT3′. The PCR-fragment was digested with *Bgl*II and *Nhe*I.

In case of MAP2 the coding sequence of MAP2 was amplified as described[27] and the PCR fragment was digested with *Xho*I and *Bam*HI and inserted into the digested pEGFP-Tub plasmid (BD Biosciences Clontech, Franklin Lakes, NJ, USA) resulting in pEGFP-MAP2. Subsequently, the SNIFP coding sequence was swapped with the EGFP sequence using *Bgl*II and *Nhe*I, resulting in pSNIFP-MAP2.

To target Histone-2B, the coding sequence of Histone-2B was amplified as described[27]. The PCR-fragment was digested using *Xho*I and *Bam*HI and ligated into the digested pEGFP-Tub plasmid, resulting in pEGFP-Hist1H2BN. Subsequently, the SNIFP coding sequence was swapped with the EGFP coding sequence using *Bgl*II and *Nhe*I, resulting in pSNIFP-HIST1H2BN.

To target peroxisomes, we generated a plasmid expressing SNIFP with the peroxisomal targeting sequence (PTS) at its C-terminus. To this end, we fused the PTS to the coding sequence of mEGFP by PCR using the primers 5′CGACGCTAGCATGGTGAGCAAGGGCG3′ and 5′AACAGGATCCCTACAGCTTGGACACTCGAGATCTGAGTCCGGACTTGTACAGCTCGTCCATGCC3′. Subsequently, this PCR-product was swapped with the coding sequence of pEGFP-Tub in pEGFP-Tub (BD Biosciences Clontech) using *Nhe*I and *Bam*HI, resulting in pEGFP-PTS. Subsequently, the coding sequence of SNIFP was swapped with the EGFP coding sequence using *Bgl*II and *Nhe*I, resulting in pSNIFP-PTS.

To target SNIFP to Vimentin, the coding sequence of SNIFP was amplified with the primers 5′TCCCCCGGGCGCCACCATGTCCCGTGACCCTCTCCCCT3′ and 5′GCGGCCGCTCATTCTTTGACTTGCACCTGTT. The PCR-product was swapped with the mKate2 coding sequence in pmKate2-Vimentin (Evrogen, Moskau, RU) using *Age*I/*Xma*I and *Not*I, resulting in pVIM-SNIFP.

For generation of a human cytokeratin-18 fusion construct, the SNIFP coding sequence was amplified with the primers 5′ACGGTACCGCGGGCCCGGGATCCACCGGTCGCCACCATGTCCCGTGACCCTCTCCCCT3′ and 5′AGCTGTGCGGCCGCTCATTCTTTGACTTGCACC TGTTC3′. The PCR-product was swapped with the TagRFP coding sequence in pTagRFP-Keratin18 (Evrogen) using *Kpn*I and *Not*I, resulting in pKrt18-SNIFP.

To target SNIFP (or mCherry) to the endoplasmic reticulum (ER), the coding sequence of SNIFP was amplified using the primers 5′CTGCAGGTCGACATGTCCCGTGACCCTCTC3′ and 5′TTCTGCGGGCCGCTTCTTTGACTTGCAC3′ (mCherry: 5′CTGCAGGTCGACATGGTGAGCAAGGGCGAGGA3′ and 5′TTCTGCGGCCGCCTTGTACAGCTCGTCCATGCCGCCGGT3′). The PCR-products were digested using *Sal*I and *Not*I and ligated into the digested pEF/Myc/ER-construct (Invitrogen). Resulting in pER-SNIFP (pER-mCherry).

To target Caveolin 1 (Cav1), the coding sequence of SNIFP was amplified using the primers 5′CGCCCGGGCGCCACCATGTCCCGTGACCCTCTC3′ and 5′GTCGCGGCCGCTTCTTTGACTTGCACCTGT3′. The PCR-product was swapped with the coding sequence of TagRFP (TagRFP-N, Evrogen) using *Age*I/*Xma*I and *Not*I, resulting in pSNIFP-N. The Sequence of Cav1 (obtained from pDONR223-CAV1[29]) was amplified using the primers 5′TCCGCTAGCATGTCTGGGGGCAAAT3′ and 5′CCGGTGGATCCCGGGCCCGCGGTATTTCTTTCTGCAAGTTGATG3′. The PCR-fragment was digested using *Nhe*I and *Bam*HI and ligated into the digested pSNIFP-N plasmid, resulting in pCAV 1-SNIFP.

To target NUP50, the coding sequence of SNIFP was amplified using the primers 5′TCCGCTAGCGCTACCGGTCGCCACCATGTCCCGTGACCCTCT3′ and 5′CACTCGAGATCTGAGTCCGGATTCTTTGACTTGCACCTGT3′. The PCR product was swapped with the coding sequence of mEmerald (Addgene plasmid #54209) using *Nhe*I and *Bgl*II, resulting in pSNIFP-NUP50.

To target the centromere protein C (CENP C), the coding sequence of CENP C (obtained from pDONR223_CENP_C[29]) was amplified using the primers 5′CAGATCTCGAGTGGCTGCGTCCGGTCTGGA3′ and 5′TCCGGTGGATCCTTAGCATTTCAGGCAACTCTCCT3′. The PCR-product was swapped with the coding sequence of tubulin (pEGFP-Tub, BD Biosciences Clontech) using *Xho*I and *Bam*HI, resulting in pEGFP-CENP C. The coding sequence of SNIFP was amplified using the primers 5′ATCCGCTAGCGCTAATGTCCCGTGACCCTCTC3′ and 5′CAGTCGACATCTGAGTCCGGATTCTTTGACTTG

CACC3′. This PCR-product was swapped with the coding sequence of EGFP (pEGFP-CENP C) using XhoI/SalI and NheI, resulting in pSNIFP-CENPC.

**Viral expression system and particle preparation**. To ensure invariant expression levels, infection with the expression system semliki forest virus (SFV) was used for comparison of mammalian cellular brightnesses (see Supplementary Table 2). The sequences of proteins of interest (SNIFP, W4.20, W4.33, W4.34) were amplified using the primers 5′TGAT TGGATCCCGGGCTCGAGCTCAAGC TTCGAATTCATGTCCCGTGACCCTCTCCCCTTCTTC3′ and 5′AGCTG TGCGGCCGCTCATTCTTTGACTTGCACCTGTTC3′. The PCR-product was swapped with the coding sequence of LA-EYFP (pSCA3-CMV-LA-EYFP-construct)[30], using BamHI and NotI, which results in pSCA3-CMV-SNIFP, pSCA3-CMV-W4.20, pSCA3-CMV-W4.33 and pSCA3-CMV-W4.34.

For virus particle generation, HEK293 cells were co-transfected (using TransIT-293 transfection reagent, Mirus Bio LLC, Madison, WI, USA) with the pSCA3-CMV-construct encoding the protein of interest and the pSCA3 helper plasmid encoding the viral structural proteins[31]. After transfection, according to manufacturer's instructions, cells were grown for four days at 37 °C under 90% humidity and 5% $CO_2$. Afterwards, cells were lysed by two freeze-thaw cycles. Cell debris was removed by centrifugation at $1700g$ for 10 min. The supernatant was centrifuged for 2 h at $47,810g$ to pellet the SFV particles. The pellet was dissolved in 150 μl TBS-5 (130 mM NaCl, 10 mM KCl, 5 mM $MgCl_2$, 50 mM Tris-HCl, pH 7.8). Before infection, particles were activated with chymotrypsin (PBS, 10 mg/ml chymotrypsin, 10 mM $MgCl_2$, 10 mM $CaCl_2$). Chymotrypsin was inactivated with aprotinin (10 mM HEPES, 10 mg/ ml aprotinin). One day after seeding in six-well plates, the cells had a density of ~70% and were infected using 5 μl of activated particles per well.

**Mammalian cell culture**. HeLa (ATCC CCl-2) and U2OS (ATCC HTB-96) cells were transfected with the respective plasmid using TurboFect Kit (Thermo Fisher Scientific, Waltham, MA, USA). Cells were cultivated in Dulbecco's Modified Eagle's Medium (DMEM-Medium: 4.5 g/L glucose, GlutaMAX, phenol red, 10% vol/vol FCS, 1 mM sodium pyruvate, 100 μg/ml streptomycin, 100 μg/ml penicillin) on coverslips (for imaging) or without coverslips (for FACS measurements) in 6-well plates at 37 °C, 90% humidity and 5% $CO_2$. Approximately, 2 h before imaging or measuring, 25 μM biliverdin was added to the medium.

**Semi-native polyacrylamide gel electrophoresis**. 4 μg purified protein dissolved in 10% sucrose, 100 mM Tris-HCl, 150 mM NaCl, pH 7.5 was loaded onto 15% polyacrylamide gels containing 0.1% sodiumdodecyl sulfate. As size standards, purified monomeric mIFP[32], rsEGFP2[23], and dimeric dTomato[26] were used. Fluorescence was detected with a homebuilt gel-recording device. To detect the NIR fluorescence, the gel was irradiated with light of 655/40 nm and fluorescence was recorded at 740/40 nm. To detect green fluorescence (rsEGFP2), the gel was irradiated with 470/10 nm and fluorescence was recorded at 525/60 nm. To detect red fluorescence (dTomato), the gel was irradiated with 545/20 nm and fluorescence was recorded at 617/37 nm. Finally, all images were overlaid for display.

**Spectral characteristics**. Absorption and emission spectra were measured on a Varian Cary 4000 UV/vis spectrometer and a Varian Cary Eclipse fluorescence spectrometer. Extinction coefficients were determined in comparison to mIFP ($55,000 M^{-1} cm^{-1}$; value based on protein concentration[32]). Absorption spectra were baseline corrected and normalized to the 280 nm peak. The spectra were corrected according to the Tyr and Trp content. All measurements were performed in replicates. $n = 2$ for W4.20, $n = 3$ for the other variants.

For direct determination of quantum yields, a Quantaurus-QY instrument (C11347, Hamamatsu GmbH Deutschland, Herrsching am Ammersee, DE) was used. All measurements were performed in Tris-buffer (100 mM Tris-HCl, 150 mM NaCl, pH 7.5). Protein solutions were excited at the respective excitation maximum and the whole emission spectrum was used for quantification of the quantum yield. All measurements were performed in replicates. $n = 3$ for W4.20 and W4.33, $n = 2$ for W4.34 and W4.35.

Measurements for pH-stability were performed using a Cytation 3 plate reader (BioTek, Winooski, VT, USA) with fluorescence excitation at 650/19 nm and detection at 700/19 nm. The values of each measurement were normalized to the corresponding maximal signal. For the different pH-conditions, the following buffers were prepared:

pH 3–5.75: 100 mM citric acid, 150 mM NaCl
pH 6–7: 100 mM $KH_2PO_4$, 150 mM NaCl
pH 7.5–8.5: 100 mM Tris, 150 mM NaCl
pH 9–9.5: 100 mM glycine, 150 mM NaCl

All measurements were performed in replicates. $n = 13$ for W4.34 and W4.35, $n = 8$ for W4.20 and W4.33.

**FACS**. For determination of cellular brightness in mammalian cells, proteins were expressed for ~20 h using the viral expression system as described above. Before the measurements, cells were incubated for ~2 h in 25 μM biliverdin, trypsinized and resuspended in PBS. All measurements were performed in replicates. $n = 12$ for W4.34 and W4.35, $n = 11$ for W4.33, and $n = 10$ for W4.20. For measuring cellular

brightness in bacterial cells, proteins of interest and heme oxygenase were expressed in BL21-AI™ cells in liquid culture with $FeCl_3$ and ALA as described above. Protein expression was induced approximately 6 h before measurement. Before the measurements, cells were washed with PBS. All FACS measurements were performed in multiple independent replicates. $n = 2$ for W1–W4; $n = 6$ for W4.20, W4.33, and W4.35, $n = 7$ for W4.34.

A FACS system (BD influx™ cell sorter, BD Biosciences) was modified by adding a 671 nm Laser (SDL-671-300T, Shanghai Dream Laser, Shanghai, CN) for excitation. The detection window was between 690 and 766 nm.

**Lifetime measurements**. The fluorescence lifetimes were measured with a homebuilt microscope that was also used for the STED measurements. The fluorescence was collected with a prototype detector (red enhanced, <50 ps jitter, Micro Photon Device, Bolzano, BZ, IT) and data acquisition was performed by a PicoHarp 300 time-correlated single photon counting (TCSPC) system (PicoQuant GmbH, Berlin, DE). The fluorescence signal was collected from transfected (VIM-SNIFP) mammalian cells, which where incubated in 25 μM biliverdin for ~2 h. The FWHM of the IRF was below 200 ps (count rate 100k), ensuring that lifetimes of >0.6 ns could be reliably measured. All measurements were performed in replicates. $n = 7$ for W4.20 and W4.35, $n = 6$ for W4.34 and $n = 3$ for W4.33.

**Imaging**. Cells were imaged in HEPES buffered DMEM (HDMEM) without phenol red, approximately 24 h after transfection and 2 h after adding 25 μM biliverdin to the cultivation medium.

Confocal microscopy (Fig. 1) was performed with a laser raster scanning microscope (Leica TSC SP8, Leica Microsystems, Wetzlar, DE), at room temperature. The microscope was equipped with a 63× NA 1.4–0.7 oil immersion objective lens. For excitation, light of 633 was used. The detection window was between 650 and 800 nm.

STED nanoscopy of cells expressing mGarnet2[14] fusion proteins was performed using an Abberior STED 775 QUAD scanning microscope (Abberior Instruments GmbH, Göttingen, DE) equipped with an UPlanSApo 100×/1.40 oil objective (Olympus, Tokio, J). The samples were excited with a 40 MHz 640 nm laser and the STED pulses were delivered by a 40 MHz 775 nm laser. For detection, a 685/70 nm filter was used. Imaging parameters are specified in Supplementary Table 3. The bleaching data was analyzed as described for SNIFP.

STED nanoscopy of cells expressing SNIFP fusion proteins was performed with a homebuilt setup, optimized for nanoscopy in the NIR optical window. Both excitation and STED light (610, 676, and 860 nm) were generated by a titanium–sapphire laser (Mai Tai—Spectra-Physics, Santa Clara, CA, USA, pulse repetition rate: 80 MHz, output pulse duration ~150 fs). The laser beam (mode locked at 860 nm) was split into two beams with a polarization beam splitter. The p-polarized beam was fed into a crystal fiber (FemtoWHITE 800, NKT Photonics A/S, Birkerød, DK) to generate a white light source for excitation and the s-polarized beam which was used for stimulated emission depletion. The white light was again split with a dichroic mirror to generate two excitation beams by passing two band pass filters (z600/20 and z676/10). Both excitation beams were sent to electro-optical modulators (EOM, LM0202, Qioptiq Photonics, Göttingen, DE) to enable switching between the two excitation channels. Each one was fed into a polarization-maintaining single-mode fiber (~10 m, PM-S405-XP, Thorlabs GmbH, Newton, NJ, USA) to clean up the Gaussian beam profile. The STED beam (mode locked at 860 nm) was fed into a long fiber (PMJ-A3AHPC, 3A-633-4/125-3-120-1-SP or PMJ-A3HPC, 3S-633-4/125-3-60-1-SP, AMS Technologies AG, Martinsried, DE) to disperse the pulsed light to ~120 or ~60 ps. This short pulse lengths were needed to accommodate for the short lifetime of SNIFP (~630 ps). Since the excitation light was generated by pumping a photonic crystal fiber as a white light source with the same laser that was used for the STED beam, synchronicity was ensured between the excitation pulses and the STED pulses. The output light was guided to pass a vortex phase plate (VPP1b, RPC Photonics, Rochester, NY, USA) to create a doughnut-shaped STED-beam pattern on the focal plane. Circular polarization was generated by a half wave plate and a quarter wave plate in the STED beam path. Dichroic mirrors were from AHF analysentechnik AG, Tübingen, DE or Chroma, Bellows Falls, VT, USA. An oil immersion objective lens (HC PL APO 100×/1.40 OIL CS2, Leica Microsystems) was used for both focusing the incoming beams (excitation and STED beams) and collecting fluorescence in combination with a tube lens (200 mm, Thorlabs). Laser beams were scanned by a "Quad Scanner" with four galvanometer scan mirrors (GSI Group, Planegg, DE). A piezo translator (NanoMax, Thorlabs) was used for coarse movement of the sample. The NIR fluorescence (700–768 nm) was collected with either a multimode fiber of 62.5μm core diameter (~1 Airy disk) and fed to an avalanche photodiode (APD) (SPCM-AQRH-13, Excelitas, Waltham, MA, USA) or pass through a 70 um pinhole and was focused to a free space prototype APD (Micro Photon Device, Bolzano, BZ, IT). The red fluorescence (620–650 nm) was fed to a multimode fiber of 62.5μm core diameter and, then, to an APD (SPCM-AQRH-13). A time-correlated single photon counting data acquisition card (DPC 230, Becker & Hickl GmbH, Berlin, DE) was used for collecting the signal from the detectors. For imaging the cells expressing VIM-SNIFP, we time-gated the detection to record only the fluorescence emitted after the depletion pulse in order to further improve the signal-to-noise ratio[33]. In order to overlay the excitation and STED beam, another beam path was generated via a removable pellicle beam

splitter (BP145B1, Thorlabs GmbH). In this case, the laser light reflected from gold nanospheres was collected for overlapping the excitation and depletion beams with a photomultiplier tube (H10723-01, Hamamatsu Photonics Deutschland GmbH). All laser power values refer to that of the back aperture of the objective lens.

To determine bleaching during STED and confocal imaging, the fluorescence signals recorded in a time series were summed up for each individual image. For data analysis, the decay of the summed up fluorescence signal was fitted by a single exponential decay function. From the fit, the background level was estimated. The decay curves shown are normalized to the background and the maximum signal. The reported decay to $1/e$ was determined on the fit. Each data point represents the average of nine measurements (STED, mGarnet2, Supplementary Fig. S7a, c), eight measurements (STED, SNIFP, Supplementary Fig. S7b, d), five measurements (SNIFP confocal 3 μW, Fig. 1j, k), six measurements (SNIFP confocal, 6 μW, Fig. 1j, k).

All imaging parameters are summarized in Supplementary Table 3.

**Reporting summary**. Further information on research design is available in the Nature Research Reporting Summary linked to this article.

## Data availability

The datasets generated and analyzed during the current study are available from the corresponding author on reasonable request. The SNIFP sequence is deposited at GenBank. Accession number: MH982583. URL: https://www.ncbi.nlm.nih.gov/nuccore/MH982583.

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

## Acknowledgments

We thank T. Gilat and S. Löbermann for excellent technical assistance, M. Andresen, T. Grotjohann, and T. Konen for help with the screening instrumentation and acknowledge J. Jethwa for proofreading the manuscript. We thank V. V. Verkhusha for sharing the plasmid pWA23h. This work was supported by the DFG funded SFB755 (project B10) and the Cluster of Excellence and DFG Research Center Nanoscale Microscopy and Molecular Physiology of the Brain (both to S.J.).

## Author contributions

M.K., H.T. and N.A.J. performed research; M.K., H.T., N.A.J., S.W.H. and S.J. analyzed data; M.K. and S.J. conceived the study. S.J. wrote the manuscript with contributions from all authors. All authors read and approved the final manuscript.

## Additional information

**Competing interests:** The authors declare no competing interests.

