## [Peer Review File · Nature Communications]

REVIEWERS' COMMENTS:

Reviewer #1 (Remarks to the Author):

In the revised manuscript Kamper et al. appropriately and satisfactorily addressed all of the points raised by reviewers. The manuscript was improved by the additional data concerning live cell STED imaging of SNIFP protein and its comparison with other far red fluorescent protein such as mGarnet2 in such type of super-resolution imaging.

Overall, I feel that the paper is now acceptable for publication in Nature Communications.

Reviewer #2 (Remarks to the Author):

Prof. Stefan Jakobs and co-workers reviewed the manuscript "Near-infrared STED nanoscopy with an engineered bacterial phytochrome" following our concerns and replying point-by-point to our doubts.

I consider the topic address in this manuscript very important, i.e., the development of bright and photo-stable fluorescent proteins in the far-red region is currently one of the most important challenge of STED microscopy, and in general fluorescence microscopy. The results showed by Prof. Stefan Jakobs and co-authors can be consider an important step towards this goal. Imaging comparison between the proposed protein SNIFP and the mGarnet2 demonstrates a slightly improvement in photo stability, but a fair comparison would require more experiments (note that excitation power is much higher for mGarnet2 than for SNIFP and in STED microscopy tuning properly the excitation power to prevent higher-order excitation is fundamental to reduce photobleaching). However, I recognise that this is out of the topic of this manuscript. The manuscript addresses a very timely topic and the review satisfy our doubts, for these reason I support the publication of this manuscript on Nature Communications.

Minor Note: Please consider review Table S3. The number of the Figures in the table is not in agreement with the text (for example Suppl. Fig.7 and 8).

Reviewer #3 (Remarks to the Author):

The authors have done a great job of addressing most of the concerns from my previous review of this manuscript. I recommend that the manuscript be suitable for publication in Nature Communications.